# Efficacy of the Newly Invented Eyelid Clamper in Ultra-Widefield Fundus Imaging

**DOI:** 10.3390/life10120323

**Published:** 2020-12-02

**Authors:** Nobuhiro Ozawa, Kiwako Mori, Yusaku Katada, Kazuo Tsubota, Toshihide Kurihara

**Affiliations:** 1Laboratory of Photobiology, Keio University School of Medicine, Shinanomachi, Shinjuku-ku, Tokyo 160-8582, Japan; n.ozawa@keio.jp (N.O.); morikiwako@keio.jp (K.M.); yusakukatada@z2.keio.jp (Y.K.); 2Department of Ophthalmology, Keio University School of Medicine, Shinanomachi, Shinjuku-ku, Tokyo 160-8582, Japan; tsubota@z3.keio.jp; 3Tsubota Laboratory, Inc., 34-304 Shinanomachi, Shinjuku-ku, Tokyo 160-0016, Japan

**Keywords:** retinal imaging, ultra-widefield fundus imaging, retinal diseases

## Abstract

Background: Ultra-widefield fundus imaging is widely used for obtaining wide angle images of the retina in one single image. Although it has a potential to obtain a wide area of retinal photographs, images are often obstructed by eyelashes or eye lids. In this study, we used a newly invented eyelid clamper, which can keep an eye open without touching conjunctiva or lid margin, to assess the efficacy in clinical use by comparing with conventional tape fixation. Methods: Ultra-widefield fundus images were captured with an ultra-widefield imaging system in 19 patients who visited to the outpatient clinic of Department of Ophthalmology, Keio University Hospital with the eyelid clamper or a conventional tape fixation. The area of imaged retinas was outlined and quantified with pixels. After obtaining images, patients answered a questionnaire. Results: The average number of pixels in total areas with the eyelid clamper or with tape fixation were 4.31 ± 0.35 and 4.32 ± 0.34 mega pixels, respectively, showing no significant difference between the groups (*p* = 0.889). The average face pain scale of the eyelid clamper was 1.13 on a scale of 0 to 5. The number of patients who did not feel any pain was nine (47.4%). Conclusions: The eyelid clamper can be applied in clinical setting and can better support obtaining sufficiently wide fundus images compared to a conventional tape fixation.

## 1. Introduction

Nowadays, ultra-widefield fundus imaging is broadly used for obtaining widefield images of the retina in one single image. It can detect lesions in the peripheral retina such as retinal tear, retinal detachment, viral retinitis, and diabetic retinopathy. It is important to describe and detect these lesions because they are vision threatening. The gold standard for detecting peripheral retinal pathology is a dilated fundus examination with scleral indentation [1]. This examination is performed by a retinal specialist thus a fundus camera has been used for the screening and the initial assessment. Fundus cameras can be used by a generalist and are less invasive than scleral indentation, only obtaining a retinal image with an angle of 30 to 45 degrees, and need multiple shots to assess the peripheral retina. Therefore, ultra-widefield fundus imaging, which provides widefield images of the retina by one single shot, has great advantages in reducing the burden to patients. In the clinical setting, Bonnay G. et al. have reported that the sensitivity of retinal detachment detection with the widefield imaging reading by a senior resident appears to be satisfactory for screening purposes [2]. Wessel M.M. et al. have demonstrated that ultra-widefield fluorescein angiography reveals significantly more retinal vascular pathology in patients with diabetic retinopathy, with 3.2 times more total retinal surface area than conventional Early Treatment of Diabetic Retinopathy Study 7 standard field fundus imaging [3].

Ultra-wide field fundus imaging also provides a variety of modalities, including fluorescein angiography, indocyanine green angiography, pseudocolor, and fundus autofluorecence [4]. For fluorescein angiography, this technology has enabled a direct angiographic view of peripheral retinal vascular anatomy, such as the nonperfusion area in diabetic retinopathy and retinal vein occlusion [5].

Although ultra-widefield fundus imaging systems have the potential to obtain up to 200 degrees of retinal images, images are often obstructed by eyelashes or eye lids (Figure 1a), even within the 7 standard field area [6]. Mackenzie P.J. et al. showed low sensitivity for lesions anterior to the equator partly due to eyelash artifact [1]. It has been reported that eyelashes blocked approximately one third of the image even though lid retractions by fingers and cotton buds were applied during the image capture [7]. Yoshinaka S. et al. compared various methods to eliminate the eyelash artifact and showed that a tape fixation and an eyelid speculum are better options than the other methods like assisting fingers or cotton buds [8]. Accordingly, tapes or eyelid speculums are generally used to keep eyes open (Figure 1b,c). Inoue M. et al. introduced a new disposable eyelid speculum which can remove the eyelash artifact completely, but they need topical anesthesia to use since the speculum touches conjunctiva and lid margin [9]. In this study, we use a newly invented eyelid clamper (CenturyArks Co., Ltd., Tokyo, Japan), which can keep an eye open without touching conjunctiva or lid margin, to assess efficacy in clinical use by comparing with conventional tape fixation.

## 2. Materials and Methods

### 2.1. Patients and Image Acquisition

This study was conducted as a prospective study, and in compliance with the Declaration of Helsinki and was approved by the institutional review boards of Keio University School of Medicine (approval number 20170235). Written informed consents were obtained from all participants. Face photos of one of the authors N.O. are shown in Figure 1 and Figure 2, and N.O. consented to the use of the photographs. Inclusion criteria consisted of patients who are 20 years old or older and have retinal diseases and visited the vitreoretinal division of Department of Ophthalmology, Keio University Hospital, Tokyo, Japan. Patients with corneal disorders, histories of ophthalmic surgery within the past 6 months, narrow-angle eyes or active infections were excluded. Ultra-widefield fundus images were captured by an ultra-widefield fundus imaging system (Optos California, Nikon, Tokyo, Japan) in 19 patients. Patients were 12 males and 7 females, and their average of age was 67.9 ± 10.6 years old. We followed the instructions by the retina specialist at the time of the appointment to determine whether pupil dilation was needed or not, and 15 patients were taken with pupil dilation while the other 4 patients were taken without pupil dilation.

### 2.2. Eyelid Clamper

The newly invented instrument called “eyelid clamper” is shown in Figure 2. The eyelid clamper consists of a plastic ring and a rubber band. The plastic ring is placed on the eyelids, and the rubber band is worn around the head. After wearing the eyelid clamper, we adjusted the position of the eyelid clamper on the eyelid so that the eye could be appropriately open. The ring part puts an appropriate pressure on the eyelid with a friction force keeping the eye open. For the control observation, a conventional tape fixation (Figure 1b) was performed on the same eye of each patient just before the eyelid clamper fixation on the same day. Assistance with fingers were applied both in the eyelid clamper group and the tape fixation group if necessary. Patients filled out a questionnaire after all examinations were finished. In the questionnaire, face pain scale in a scale of 0–5 during and after applying the eyelid clamper, locations of the pain, and additional comments for wearing the eyelid clamper were asked. For face pain scale, we used Wong-Baker FACES^®^ Pain Rating Scale (Wong-Baker FACES Foundation, Oklahoma City, OK, USA) with permission.

### 2.3. Image Analysis

The area of the imaged retina was outlined manually by using Adobe Photoshop software (Photoshop CC version 19.1.6, Adobe Systems Incorporated, San Jose, CA, USA) and quantified with pixels [10]. It is known that images are affected differently in each quadrant. For example, eyelash artifacts are more likely to be seen in inferior periphery [1]. Therefore, the images were divided into superior, inferior, nasal, and temporal quadrant using two 45-degree lines and each quadrant was also quantified.

### 2.4. Statistical Analysis

Statistical analysis was performed using SPSS 25 (IBM, New York, NY, USA). Paired sample *t*-tests was used to compare pixels of visible retinas of the patients with tape fixation and with the eyelid clamper. We did not use conventional metal eyelid speculums in this study.

## 3. Results

The ultra-widefield fundus imaging was successfully performed with both conventional tape fixation and the eyelid clamper (Figure 3). The average age of the patients was 67.9 ± 10.6 years old, 17 patients had phakic eyes, 13 patients had cataracts, and 2 patients had intraocular lens. The average refraction (spherical equivalent) was −3.13 ± 4.75 diopters. The main diseases of the patients were age-related macular degeneration (7), retinal vascular occlusion (4), diabetic retinopathy (3), retinal tear (1), viral retinitis (1), and inherited retinal degeneration (1).

The average number of pixels in the total areas of the conventional tape fixation and the eyelid clamper were 4.32 ± 0.34 and 4.31 ± 0.35 mega pixels, respectively (Figure 4a). There was no significant difference between the two groups (*p* = 0.889). In the nasal quadrants, the average area of the visible retina in the eyelid clamper group was significantly larger than the eyelid clamper group (*p* = 0.0003, Figure 4b). On the other hand, the average area was significantly larger in the tape group compared to the eyelid clamper group in the temporal quadrants (*p* = 0.0087, Figure 4c). There was no significant difference in superior and inferior quadrants (Figure 4d,e).

The average face pain scale during and after wearing the eyelid clamper were 1.13 ± 1.35 and 0.10 ± 0.31, respectively (Table 1). The location of the pain was on both temples in three patients, the lower eyelid of the unexamined eye in one patient, the lower eyelid of the examined eye in one patient, and around the nose in one patient, and all of these were caused by the rubber band. The pain on the examined eye was found in four patients; two of them were from dryness of the eye, and one patient had a sore eye.

## 4. Discussion

In this study, we evaluated the efficiency of the newly invented eyelid clamper and found that the average area of the visible retina was almost equal between the conventional tape fixation group and the eyelid clamper group. In some cases, with the eyelid clamper, an obstruction by the nasal part of the plastic ring was seen in the temporal area of the fundus image (Figure 3d) which might result in smaller visible retinal images in the temporal quadrant (Figure 4c). On the other hand, the tape group showed significantly smaller visible retinal images in the nasal quadrant, partly because examiners had to support the eyelid to open with their fingers from the temporal side of the patient, even with tape fixation (Figure 3d).

In order to avoid eyelash artifacts, tape fixation, examiner’s assistance with cotton swab, and eyelid speculum have been used. Comparisons of these types of assistance are summarized in Table 2. With tape fixation, it is known in a clinical practice that some people feel pain when removing it, and there is the risk of epidermal peeling. Yoshitake S. et al. reported that the patients complained of dryness of the eye when keeping eyes open with tape or eyelid speculum and had sore eyes when using the topical anesthesia [8], although Inoue M. et al. reported that use of a disposable eyelid speculum with topical anesthesia removed the eyelash artifacts completely [9]. Thus, use of eyelid speculum and topical anesthesia could decrease the efficiency of the examination as well as patient comfort during the examination [11]. In addition, topical anesthesia may cause side effects such as allergies and corneal erosions. The eyelid clamper could be applied without topical anesthesia meaning that it has some advantage in efficiency and patient comfort. It is probably because the eyelid clamper does not have contact with the conjunctiva or cornea. The eyelid clamper can also have advantages in procedures like fluorescein angiography that would require longer period of eye exposure and procedures with children since they often cannot not tolerate ocular anesthesia and cooperate with the examination.

Some patients complained of symptomatic pain. Most of the pain which patients complained of was due to the tightening of the rubber band. Locations of the pain were around the nose, the zygomatic bone, and the temple, mainly on the opposite side of the eye examined; thus, an improvement in design of the band is required to reduce the pain. The other patients complained of the pain on the cornea such as sore eyes and dryness sensation. One of these patients had the past medical history of atopic dermatitis suggesting that patients who have inflammation on ocular surface may be needed to wear the eyelid clamper carefully.

There may be some possible limitations in this study. First, we only examined corneal lesions after obtaining ultra-widefield fundus imaging; thus, we could not describe the adverse effect of the eyelid clamper to the other parts. Still, we believe that the eyelid clamper would not cause serious adverse effects on the cornea. Second, time efficiency using the eyelid clamper is unknown because we did not measure the duration of the examination. Some patients complained that wearing and adjusting eyelid clamper were bothersome. This might be improved by a learning effect of examiners. Finally, whether pressure on eyelids generated by the eyelid clamper can affect intraocular pressure is unknown. If there is no effect on intraocular pressure, the device can also be applied for the measurement of intraocular pressure with an applanation tonometer.

In conclusion, the eyelid clamper can be applied in a clinical setting and can support the acquisition of sufficient wide fundus images, the same as with conventional tape fixations. We believe that the eyelid clamper is the best option for patients which can be used without anesthesia and any pain after removing it, and can obtain effective images, although further improvement is needed to reduce the pain and to enlarge the area of images.

## Figures and Tables

**Figure 1 life-10-00323-f001:**
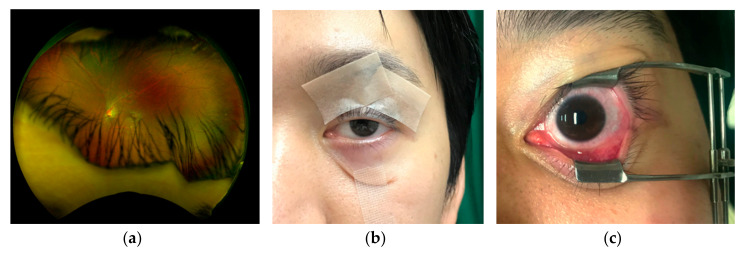
Conventional fixations of eyelid to capture widefield retinal images. (**a**) A representative fundus photograph obstructed by eyelash and eyelid. (**b**) Tape fixation. (**c**) Metal eyelid speculum fixation.

**Figure 2 life-10-00323-f002:**
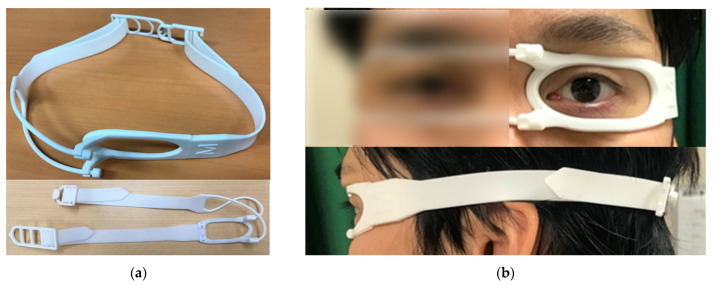
The eyelid clamper. (**a**) The entire external form of the eyelid clamper (**upper**). The adjustable and dividable band (**lower**). (**b**) An image of eyelid fixation with the front (**left**) and the side (**right**).

**Figure 3 life-10-00323-f003:**
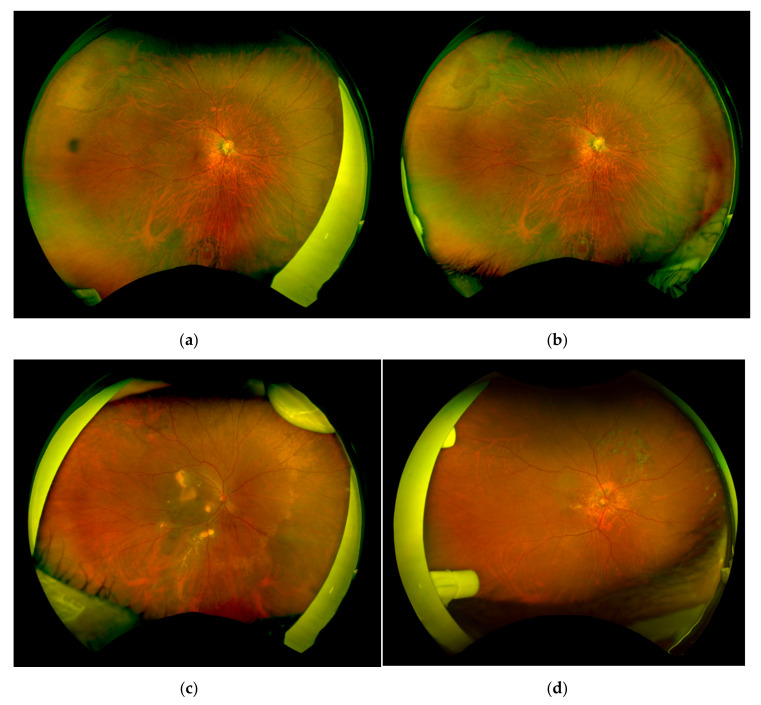
Ultra-widefield fundus photographs. Representative retinal images with a tape fixation (**a**) and the eyelid clamper (**b**). Representative images obstructed by examiner’s finger to support insufficient fixation with tape (**c**), or parts of the eyelid clamper (**d**).

**Figure 4 life-10-00323-f004:**
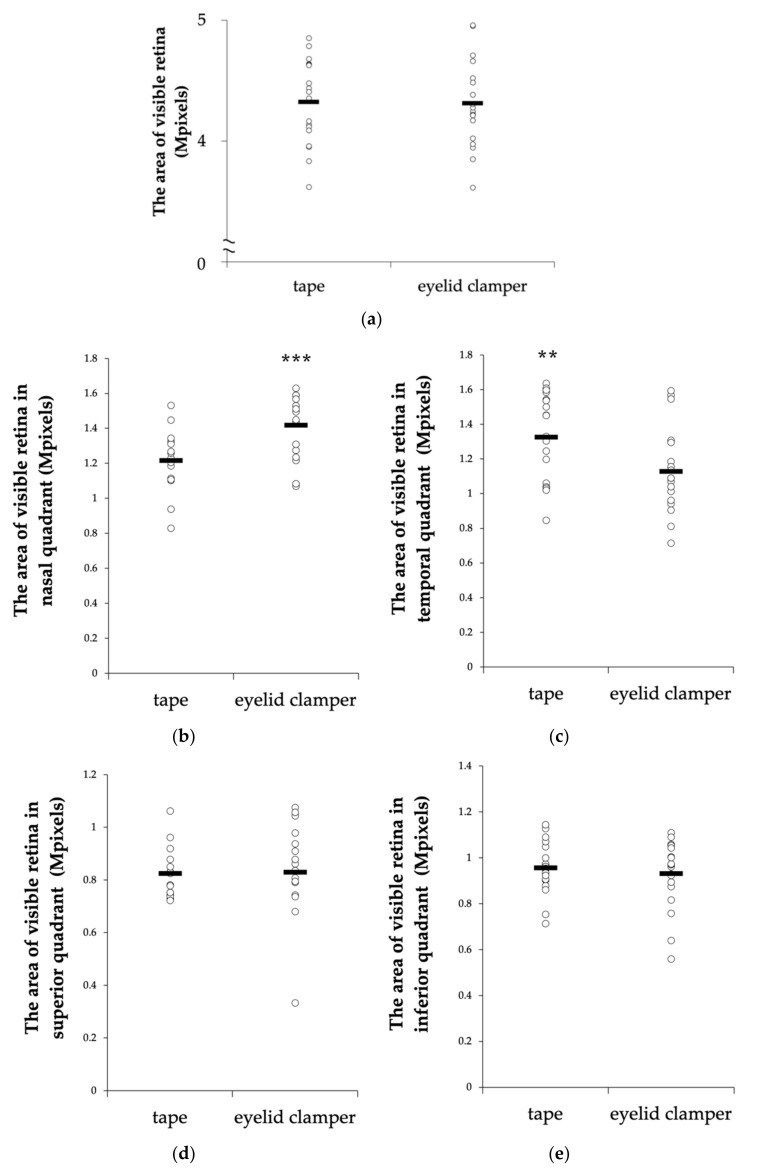
Quantification of the visible retinal areas. The visible area is quantified in the total areas (**a**), nasal quadrants (**b**), temporal quadrants (**c**), superior quadrants (**d**), and inferior quadrants (**e**). ** *p* <0.01, *** *p* < 0.001.

**Table 1 life-10-00323-t001:** Results of the questionnaire.

Items	Results
Face pain scale while wearing the eyelid clamper	1.13 ± 1.35
Face pain scale after wearing the eyelid clamper	0.10 ± 0.31
Location of the pain	Both temples in 3 patients
	The lower eyelid in 2 patients
	Around the nose in 1 patient
	The examined eye in 4 patients

**Table 2 life-10-00323-t002:** Summary of each fixation method.

	Tape	Eyelid Clamper	Conventional Eyelid Speculum
Topical anesthesia	-	-	+
Pain	Caused by adhesion	Mostly by pressure of the band	Cannot be done without anesthesia

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
