# Peer review of "Efficacy of the Newly Invented Eyelid Clamper in Ultra-Widefield Fundus Imaging"

_life, 2020, doi:10.3390/life10120323_

Round 1

Reviewer 1 Report

The manuscript (#life-977411, by Ozawa N, et al.) is a study probably to examine the efficacy and safety of the eyelid clamper the authors newly invented on ultra-widefield fundus imaging. Intuitively, the eyelid clamper would have advantages over other methods: tape fixation and speculum fixation. Accordingly, I recommend this manuscript for publication if the authors adequately respond to my concerns listed below.

  • It should be clearly stated whether this study was a prospective or not. Plus, if it was a prospective study, the rationale by which sample size of 19 was determined.
  • From the mentions in Introduction and Results, the aim of this study surely lies in comparison between the eyelid clamper fixation and other fixation methods. If so, please amend the descriptions of Background in Abstract and Introduction of the main text.
  • In Abstract, the results of the questionnaire should be provided in Results.
  • In Conclusions of Abstract, please make it clear what “conventional fixation” means: finger, cotton swab, tape, or speculum.
  • In Introduction, the problems related to tape fixation and speculum fixation should be explained here with references so that the authors can understand the need of a novel fixation method.
  • In 2.1 Patients and image acquisition, the inclusion and exclusion criteria should be shown. In addition, the information should be provided on whether images were acquired through a dilated pupil or not.
  • In 2.1 Patients and image acquisition, the following should be clearly described: (1) did the same patient undergo three types of ultra-widefield fundus imaging: tape fixation, speculum fixation, and eyelid clamper fixation?; (2) if (1) is true, were examinations performed on the same day?; (3) if (2) is true, how did the authors decide the order of examinations (randomly or using prefixed order)?
  • In 2.2 Eyelid clamper, the detailed information on the questionnaire should be given: the number of questions, prepared interrogative sentences, prepared alternatives for answer, etc.
  • On page 3, line 102: I do not think that the data of questionnaire was fully presented (e.g., pain scale in tape fixation).
  • In 3. Results, more detailed patients’ demographic data should be provided including, lens status, axial length, refraction, and vitreoretinal disorders which would have some influence on pixel numbers.
  • On page 5, lines 127-133: although the authors discussed about pain in this paragraph, most of the data mentioned here is not presented in Results section.

Reviewer 2 Report

This is an interesting article presenting an instrument that might be useful in fundus evaluation using ultra-widefield imaging .

Minor revisions are suggested before possible publication.

There seems to be a discrepancy between what is said in lines 97-100 and what is shown in figures 4b and 4c, please revise.

Moreover, an English revision might be needed (“average age” line 21, in line 32 the word “wide” is used three times; “need-needs” line 39, “to support to keep” line 55, “has been” line 119, “complained the pain on the cornea” lines 130-131, “clumper” line 151…). Reference 1 needs to be put between brackets in line 55.

Reviewer 3 Report

Review report:

The objective of the study was to assess the efficacy of the new eyelid clamper in daily clinical use. The authors quantified the number of pixels showing no statistical significance between the new eyelid clamper and the traditional clampers.

The article is interesting and well structured, although there are several areas of improvement to become the article interesting and clearer for the reader:
1. Introduction: please develop more about eyelid traditional clamps and the issues around it (create the need for this new invention), so we can understand later the advantages of the new eyelid clamp.

2. Material and methods:
2.1 - Please specify N.O.
2.2 – The authors state that a questionnaire was performed to patients and this included: face pain, locations of the pain, additional comments. Please create a table with the questions performed to patients and analyze the questions. The study studies efficacy however analyzing the patient’s opinion, patients' comfort and the overall acceptance of this new eyelid clamper by the patients (which ultimately may lead to a better-preformed exam) would be interesting.
2.3 – Please clarify to the reader what did you look for when analyzing each quadrant.
2.4 – Please clarify if the traditional clamper used to test the hypotheses was tape fixation only or also the eyelid speculum. Confirm if there was or not help from the assistant with fingers during this trial.

3. and 4. - Results and discussion – Clarify the efficacy results and include an analysis of the patient's questionnaire.

Inline 101, you stated “data not shown”, please show the missing data.

The information provided on the data collected can be improved.

My advice would be to include more information which would turn a very interesting article, simple and of easy reading:

  1. Information about patient characteristics,
  2. Information about the questionnaires performed to the patients (not only the questionnaire but also its analysis and interpretation),
  3. Information about data collected and not shown
    Which would turn the paper more interesting.

Also, show:

  1. Through a table (for example) the differentiation of this new eyelid clamper and traditional eyelid clampers;
  2. The unmet need for this new eyelid clamper and what are the gaps this new eyelid clamper fills;
  3. Finally include a paragraph why the authors believe:
    1. It is the best option for patients in terms of efficacy and quality of the exams obtained and
    2. In terms of the patient's comfort and preference.

Round 2

Reviewer 1 Report

The manuscript is well revised and now I recommend it for publication.

Reviewer 3 Report

Dear authors,

Thank you for your corrections and for accepting my suggestions.

No further comments.

This manuscript is a resubmission of an earlier submission. The following is a list of the peer review reports and author responses from that submission.